# Function-Preserving Gastrectomy for Early Gastric Cancer

**DOI:** 10.3390/cancers13246223

**Published:** 2021-12-10

**Authors:** Yoshihiro Hiramatsu, Hirotoshi Kikuchi, Hiroya Takeuchi

**Affiliations:** 1Department of Surgery, Hamamatsu University School of Medicine, 1-20-1 Handayama, Hamamatsu 431-3192, Japan; hiramatu@hama-med.ac.jp (Y.H.); kikuchih@hama-med.ac.jp (H.K.); 2Department of Perioperative Functioning Care and Support, Hamamatsu University School of Medicine, 1-20-1 Handayama, Hamamatsu 431-3192, Japan

**Keywords:** gastric cancer, function-preserving, sentinel node

## Abstract

**Simple Summary:**

For patients with early gastric cancer (EGC), a good prognosis is achieved by conventional standard gastrectomy with radical lymphadenectomy. However, postgastrectomy syndrome is often inevitable and results in decreased quality of life (QOL). To improve patients’ QOL, proximal gastrectomy instead of total gastrectomy and pylorus-preserving gastrectomy instead of distal gastrectomy have been widely accepted as function-preserving gastrectomies. Recently, personalized, minimized gastrectomy with sentinel node navigation surgery has been developed and is expected to be an ideal treatment option for patients with EGC. Herein, we review the indications, surgical techniques, and postoperative outcomes of function-preserving gastrectomy.

**Abstract:**

Recently, minimally invasive (endoscopic or laparoscopic) treatment for early gastric cancer (EGC) has been widely accepted. However, a standard gastrectomy with radical lymphadenectomy is generally performed in patients with EGC who have no indications for endoscopic resection, and postgastrectomy dysfunction is one of the problems of standard gastrectomy. Function-preserving gastrectomy, such as proximal gastrectomy and pylorus-preserving gastrectomy, can be considered when attempting to preserve the patient’s quality of life (QOL) postoperatively. In addition, sentinel node navigation surgery for EGC has been applied in clinical practice in several prospective studies on function-preserving personalized minimized gastrectomy. In the near future, the sentinel lymph node concept is expected to form the basis for establishing an ideal, personalized, minimally invasive function-preserving treatment for patients with EGC, which will improve their postoperative QOL without compromising their long-term survival. In this review article, we summarize the current status, surgical techniques, and postoperative outcomes of function-preserving gastrectomy for EGC.

## 1. Introduction

Recently, minimally invasive approaches, such as endoscopic treatment or laparoscopic gastrectomy (including robot-assisted surgery), for early gastric cancer (EGC) have gained wide application in clinical practice [1]. However, standard gastrectomy with radical lymphadenectomy is generally performed for patients with EGC who have no indications for endoscopic submucosal dissection (ESD), but postgastrectomy dysfunction is one of the problems of standard gastrectomy. Due to the low incidence of lymph node metastasis and the excellent prognosis in EGC, function-preserving gastrectomy, with an adequate range of gastric resection and minimal lymphadenectomy, could improve the patient’s quality of life (QOL) [2]. Proximal gastrectomy (PG) and pylorus-preserving gastrectomy (PPG) are examples of function-preserving gastrectomies that can be performed in patients with EGC. PG is an alternative to total gastrectomy (TG) for patients with EGC located in the upper portion of the stomach, whereas PPG is an alternative to distal gastrectomy (DG) for patients with EGC located in the middle portion of the stomach. In addition, sentinel node navigation surgery (SNNS) for EGC has been applied in clinical practice in several prospective studies on function-preserving, personalized, minimized gastrectomy [3,4]. The concept of the sentinel lymph node (SN) is expected to be useful in selecting personalized, function-preserving surgery for patients with EGC. In this review article, we summarize the current status of function-preserving gastrectomy for EGC.

## 2. Literature Search

A systematic literature search of the PubMed database was carried out until August 2021. The search was limited to studies published in English. The search terms were as follows: “gastric cancer or gastric neoplasm” and “pylorus-preserving or pylorus preserving” or “function-preserving or function preserving” or “proximal gastrectomy” or “sentinel node or sentinel lymph node” or “local resection or local gastrectomy.”

The inclusion criteria for the evaluation of postoperative weight change were as follows: (1) studies of patients with pathologically confirmed gastric cancer, (2) comparisons of PG to TG, (3) comparisons of PPG to DG, (4) studies of patients who underwent local resection of the stomach, and (5) the revealed adequate data of postoperative weight changes. Exclusion criteria were as follows: (1) overlapping publications or duplicated data and (2) reviews, case reports, comments, and conference abstracts.

## 3. Proximal Gastrectomy (PG)

The incidence of proximal gastric cancer has increased in recent years [5,6]. Previously, extensive resection, such as TG with extensive lymph node dissection, was performed even for patients with relatively early-stage cancer. PG is an alternative to TG for EGC located in the upper third of the stomach [7]. Thus, PG is widely performed as a function-preserving gastrectomy for early proximal gastric cancer.

### 3.1. Indication for Proximal Gastrectomy

PG is performed in patients diagnosed preoperatively with cT1N0M0 primary gastric cancer in the upper third of the stomach and whose remnant stomach is more than half the original size [7]. This indication aims to maintain both the oncological curability and functional capacity of the remnant stomach.

Patients who underwent PG had a good prognosis with a 5-year overall survival rate of 94–97%, and some observational studies showed no difference in the recurrence and long-term survival rates associated with PG when compared with TG [7,8,9]. In patients with EGC located in the upper stomach, metastasis rates and therapeutic indices at lymph node station Nos. 4d, 5, and 6 are remarkably low, even in proximal advanced gastric or esophagogastric junctional cancers [10,11,12]. Although there are concerns that lymph node dissection at station No. 3b might be inadequate in an attempt to preserve the distal lymph region of the lesser curve of the stomach, the frequency of metastasis of EGC located in the upper stomach to the lymph region has been reported to be very low [13].

Although retrospective observational studies on PG for gastric cancer have been conducted, the level of evidence is generally not strong because there are no prospective randomized trials comparing the outcomes of PG with those of other surgical procedures involving TG [7,8,9,13,14,15]. PG is weakly recommended as a therapeutic option for upper EGC in the Japanese Gastric Cancer Treatment Guidelines [7].

### 3.2. Surgical Strategy of Proximal Gastrectomy

#### 3.2.1. Lymphadenectomy in Proximal Gastrectomy

D1 lymph node dissection includes lymph node station Nos. 1, 2, 3a, 4sa, 4sb, and 7; station Nos. 8a, 9, and 11p are included in D1+ lymphadenectomy [7]. Recently, station No. 11d was included in D2 dissection in the new edition of the Japanese Gastric Cancer Treatment Guidelines. During lymphadenectomy, the right gastric, right gastroepiploic, and infra-pyloric arteries are routinely preserved, and the hepatic and pyloric branches of the vagus nerve are also commonly preserved.

#### 3.2.2. Reconstruction after Proximal Gastrectomy

In PG, the rate of complications, such as reflux esophagitis and anastomotic stenosis, was markedly higher compared with TG [16,17,18,19]. Reconstruction after PG requires technical ingenuity to overcome this problem. Although the optimal method remains controversial, esophagogastrostomy (EG), jejunal interposition (JI), and double-tract reconstruction (DT) are methods that have been widely accepted for reconstruction following PG [12,16,20,21,22,23,24,25,26,27].

EG is considered a simple, physiological reconstruction method because there is only one anastomotic site; however, the frequency of postoperative reflux esophagitis may be high. In recent years, good outcomes from novel reconstruction methods, such as EG with fundoplication [12,16], the double-flap technique (DFT) [20,21], and side overlap fundoplication by Yamashita (SOFY) [22] devised to prevent gastroesophageal reflux, have been reported, and the application of these reconstruction methods is becoming widespread. DFT is an antireflux procedure during EG, and consists of a unique, multistep process involving creating an H-shaped seromuscular double-flap, fixing the posterior wall of the esophagus and the anterior wall of the remnant stomach, end-to-side anastomosis between the esophagus and remnant stomach, and closing the double-flap. DFT is performed using hand-sewn techniques, but modified procedures have been developed using surgical staplers [20]. SOFY can be relatively easy to perform with laparoscopic surgery using a linear stapler and may overcome postoperative reflux and anastomotic stenosis [22].

Reconstruction methods using the jejunum, such as JI and DT, are also used to reduce gastroesophageal reflux. JI is a more physiological reconstruction method that uses the jejunum in form; however, it is slightly complicated. In the era of laparoscopic surgery, the frequency of JI has decreased due to its complexity, and DT, which is easier to perform laparoscopically, has become more common [28,29]. DT is generally performed as follows: an esophagojejunostomy is performed with a circular stapler or linear stapler; side-to-side gastrojejunostomy, 8–10 cm below the esophagojejunostomy, is performed using a linear stapler; and a jejunojejunostomy is performed with a circular stapler, linear stapler, or hand-sewn sutures. This method is similar to a Roux-en-Y reconstruction after TG. In DT, there are two food routes: one through the remnant stomach and the other reaching the jejunum on the anal side without the remnant stomach. Ideally, the former is the main route of food, and the latter is the escape route when the remnant stomach is full. Therefore, to facilitate the passage of food from the jejunum to the remnant stomach, it is recommended to attach and secure the remnant stomach or the jejunum between the esophagus and the remnant stomach to the incision part of the lesser omentum.

### 3.3. Surgical Outcomes of Proximal Gastrectomy

PG has several advantages over TG in patients with proximal EGC, including surgical outcomes and long-term nutritional status [8,9,12,17,18]. A meta-analysis of 2036 patients in 18 studies showed that PG was potentially superior to TG in terms of operative time, intraoperative blood loss, and postoperative weight loss [18]. As shown in Table 1, the rate of change in body weight after PG and TG was 86.4–90.4% and 81.6–87.5%, respectively. In addition, some observational studies have shown that postoperative reduction in the levels of serum hemoglobin, ferritin, and vitamin B12 is milder after PG than after TG [8,9,17,18]. Regarding changes in nutritional indicators such as total protein, albumin, total cholesterol, and total lymphocyte count, there are variations depending on the reconstruction methods and studies.

The incidence of complications after PG, such as anastomotic leakage, bleeding, and pancreatic fistula, is considered the same as after TG, except for anastomotic stenosis and reflux esophagitis [16,17,18,19]. The incidence of anastomotic stricture was reported to range from 6.1–27.5% in EG [8,15,16,17,30,31], 4.7–9.0% in DFT [20,21,32], 0% in SOFY [22], and 3.3–9.1% in esophagojejunostomy [9,24,25,33]. The incidence of reflux esophagitis, evaluated as grade B or higher according to the Los Angeles classification, was reported to range from 9.8 to 32.3% in EG after PG [19,27]. It has been reported that reflux esophagitis can be reduced to 5.0–9.1% by EG with anti-reflux specifications [15,17,30,31], 0–6.0% by DFT [20,21], 7.1% by SOFY [22], and 0–5.0% by reconstructions using the jejunum, such as JI and DT [19,24,25,34].

A multicenter study using the Postgastrectomy Syndrome Assessment Scale-45 (PGSAS-45) to investigate postoperative long-term QOL, reported that PG was better than TG in terms of needing additional meals, experiencing diarrhea, and dumping symptoms [11].

## 4. Pylorus-Preserving Gastrectomy (PPG)

PPG is a surgical procedure for EGC located in the middle third of the stomach; it is generally thought to offer several advantages over DG with Billroth I reconstruction, especially in the incidence of dumping symptoms and bile reflux gastritis.

### 4.1. Indication for Pylorus-Preserving Gastrectomy

PPG is a type of function-preserving gastrectomy for EGC (without lymph node metastasis) located in the middle third of the stomach when the distance between the distal tumor border and pylorus exceeds 4 cm [7,37,38,39,40]. It is usually not performed in the elderly (over 75 years of age) or patients with hiatal hernia and reflux esophagitis [40].

In PPG, there are concerns that in a bid to preserve the pyloric cuff, the dissection of lymph nodes in stations 5 and 6 is inadequate. However, it was reported that the possibility of micrometastasis to the lymph nodes in stations 5 and 6 might be negligible for EGC located in the middle portion of the stomach [41,42]. It was also reported that metastasis to lymph nodes along the infrapyloric artery (IPA), namely those at station No. 6i, was not observed in early middle-third gastric cancer, suggesting that the dissection of lymph node station No. 6i may be unnecessary in PPG and that the IPA can be preserved [43].

The 5-year overall survival rate of PPG was reported to range from 96.3% to 98.0%. [37,40,44]. A multicenter propensity score-matched cohort study comparing PPG with conventional DG revealed the oncological safety of PPG for clinical T1N0 EGC in the middle portion of the stomach by analyzing the 5-year overall and 3-year relapse-free survival rates [45].

All studies comparing the outcomes of PPG to those of DG are observational, except for one multicenter prospective randomized trial with a small sample size [46], and the overall evidence of the superiority of PPG over DG is not very strong. PPG is weakly recommended as a therapeutic option for central EGC in the Japanese Gastric Cancer Treatment Guidelines [7].

### 4.2. Surgical Strategy in Pylorus-Preserving Gastrectomy

#### 4.2.1. Lymphadenectomy in Pylorus-Preserving Gastrectomy

D1 lymph node dissection includes lymph node station Nos. 1, 3, 4sb, 4d, 6, and 7; station Nos. 8a and 9 are included in D1+ lymphadenectomy, according to the Japanese Gastric Cancer Treatment Guidelines [7]. However, in PPG, dissection of station No. 6i can be forgone [7,43], and the IPA and infra-pyloric vein are preserved. During lymph node dissection, the preservation of the nerve supply and blood flow to the pyloric antrum is important for maintaining pyloric function. Therefore, the proximal parts of the right gastric artery and vein and the right gastroepiploic artery and vein were preserved; the right gastric artery and vein were transected at a point distal to its first branch, while the right gastroepiploic artery and vein were transected at a point distal to its infra-pyloric branches. The hepatic and pyloric branches of the vagus nerve were generally preserved. It has been reported that the preservation of blood flow and vagal branches of the antrum is essential for preventing gastric stasis [38,41,46,47,48].

#### 4.2.2. The Length of the Pyloric Cuff in Pylorus-Preserving Gastrectomy

In PPG, the length of the pyloric cuff is also important to prevent pyloric dysfunction. It was reported that preserving a 2.5 cm pyloric cuff was associated with a lower incidence of postoperative stasis than a 1.5 cm cuff, which might reduce gastric wall motility due to severe postoperative edema of the pyloric cuff [49]. Several studies have shown that retaining a pyloric cuff measuring over 3 cm does not affect the incidence of postoperative stasis [47,50]. Therefore, the length of the pyloric cuff retained most often measures between 3 and 4 cm [6,27,28,38,48].

#### 4.2.3. Reconstruction after Pylorus-Preserving Gastrectomy

For gastro-gastro anastomosis after PPG, hand-sewn anastomosis, especially layer-to-layer anastomosis, is the most widely used technique [48,51]. Even in laparoscopic surgery, this anastomosis is often performed extracorporeally using hand sutures [51]. According to an assessment using the PGSAS-45 of postgastrectomy symptoms after PPG, it was reported that the nausea score in patients who underwent hand-sewn anastomosis was significantly lower than that in those who underwent stapled anastomosis [51]. In totally laparoscopic PPG, gastro-gastrostomy is performed using a linear stapler with a delta-shaped anastomosis [52]. Although delta-shaped methods are easy and safe, they require a great deal of attention to prevent deformation, twisting, and stenosis around the anastomosis. Recently, a novel intracorporeal end-to-end gastro-gastrostomy method, called the piercing method, has been reported, and this method can maintain a wide circular shape with the antral cuff maintained circumferentially [53].

### 4.3. Surgical Outcomes of Pylorus-Preserving Gastrectomy

PPG has several advantages over DG for patients with EGC in the middle third of the stomach, including surgical outcomes and long-term nutritional status [27,39,40,51,54,55]. A meta-analysis of 4871 patients in one randomized clinical trial (RCT) and 20 non-RCTs showed that PPG was potentially superior to DG in terms of lower incidence of anastomotic leakage, dumping syndrome, gastritis, and bile reflux, and better recovery of total protein, albumin, hemoglobin, and weight [55]. As shown in Table 2, the changes in body weight following PPG and DG were 93.1–97.0% and 90.0–95.0%, respectively. No difference was found between the groups regarding operative time, blood loss, and overall complications [55].

The cumulative incidence of gallstones was also reported to be lower in PPG than in DG [31]. The preservation of the hepatic branch of the vagus nerve can be recommended in PPG to reduce postoperative gallstone formation [56]. The benefit of preserving the celiac branch of the vagus nerve is controversial, and reports on its effectiveness are two-sided [54,57].

The length of the pyloric cuff is as described in the previous section, but the size of the proximal remnant stomach affects the change in weight and postoperative QOL [51]. This study suggests that preserving a sufficient proximal stomach is recommended for function-preserving PPGs.

On the other hand, PPG is associated with a long hospital stay, decreased lymph node collection, and delayed gastric emptying [55]. In particular, delayed gastric emptying, which is found postoperatively in 6–8% of patients who undergo PPG, is a characteristic complication that should be considered [37,38,39]. Recently, a multicenter RCT showed that postoperative pyloric stenosis was significantly more frequent in the PPG than DG groups (7.3% vs. 1.6%, *p* = 0.026) [58]. Gastric stasis is a particularly important issue for patients after PPG, and maintaining pyloric blood blow, preserving the vagal branches, and keeping a sufficient length of pyloric cuff are essential to prevent this problem.

## 5. Sentinel Node Navigation Surgery

The SN is the first lymph node to receive lymphatic flow from the primary lesion and is regarded as the first possible node of metastasis [64]. If the evaluation for metastasis to the SN is confirmed by histological examination as negative, all regional lymph nodes can be predicted to be negative for cancer metastasis. Therefore, unnecessary radical lymph node dissection can be avoided by personalized surgical treatment using SNNS [2,3,4].

### 5.1. Indication for SNNS for EGC

A meta-analysis of 2128 patients in 38 studies showed that the detection rate of the SN was 94%, and the accuracy of SN diagnosis was 92% [65]. A prospective multicenter clinical trial (UMIN ID: 000000476) was conducted by a research group of the Japanese Society of SNNS to demonstrate the feasibility of SN mapping and biopsy in EGC [4]. In this study, the SN detection rate and diagnostic accuracy were 97.5% and 99.0%, respectively. This study demonstrated that metastasis-positive lymph nodes were confined to the SN basin in clinical T1N0 EGC with a major axis of 4 cm or less. Following this study, a prospective multicenter trial on personalized laparoscopic function-preserving gastrectomy combined with SN biopsy was performed to assess the long-term prognosis and QOL of patients after surgical treatment (UMIN ID: 000014401) [2]. The indication for this trial was clinical T1N0M0 EGC with a single primary lesion of less than 40 mm, without an indication for ESD. The SENORITA (SEntinel Node ORIented Tailored Approach) trial, a multicenter RCT on SNNS and standard gastrectomy for EGC, was conducted in Korea [66]. The inclusion criterion for the SENORITA trial was cT1N0M0 EGC less than 3 cm in diameter and no indication for ESD. These studies are in a state of observation to evaluate long-term prognosis after the accumulation of cases is completed.

Recently, the results of a multicenter retrospective cohort study of SN mapping in patients with EGC after ESD were reported [67]. The feasibility of SN mapping based on the SN basin concept has been clarified even in patients with EGC who have previously undergone endoscopic resection. Following this study, a prospective trial was planned to validate the SN concept in patients with EGC after ESD [67]. Moreover, a multicenter prospective trial of laparoscopic sentinel basin dissection after ESD for EGC is ongoing in Korea (SENORITA 2 trial) [68]. SNNS may also be indicated if additional surgical resection is recommended after ESD. A combination of ESD and laparoscopic SN basin dissection is a potential novel, whole-stomach-preserving, minimally invasive treatment for patients with EGC with metastasis-negative SN.

Although there are only a few retrospective studies with small sample sizes, the prognosis of patients treated with SNNS has been reported to be good, with a 5-year overall survival rate of 98.0–98.5% [69,70,71]. SNNS may be oncologically safe in EGC, but we need to wait for the results of ongoing Japanese and Korean trials.

SNNS for patients with EGC is performed as a clinical trial rather than a standard treatment and, thus, is not mentioned in the Japanese Gastric Cancer Treatment Guidelines [7].

### 5.2. Surgical Strategy for SNNS for EGC

#### 5.2.1. SN Mapping Procedures

The dual-tracer method using both radioactive colloids and blue or green dye is considered the standard procedure in SNNS for patients with EGC [3,4]. Isosulfan blue, indocyanine green (ICG), and patent blue are the most frequently used dye tracers, and these dyes are useful for visualizing lymphatic flow, even in laparoscopic surgery. ICG fluorescence imaging is useful for visualizing lymphatic flow during SN mapping [2]. Tecnetium-99m colloids (tin, sulfur, and antibody sulfur) are the preferred radioactive tracers.

The day before surgery, the radioactive colloid solution is endoscopically injected into the submucosal layer around the primary lesion using an endoscopic puncture needle. The ICG solution is also injected in the same manner into the submucosal layer surrounding the primary lesion to function as a tracer during surgery. The lymphatic vessels and lymph nodes are stained green and visualized within 15 min of the injection of the ICG solution. Simultaneously, ICG fluorescence imaging using near-infrared technology, such as VISERA ELITE II (Olympus, Tokyo, Japan) and Firefly of the da Vinci Surgical System (Intuitive, Sunnyvale, CA, USA), enabled easy visualization (Figure 1). Subsequently, the radioactive SN is detected using a gamma probe. Lymph nodes with radioactivity ten times higher than the surrounding tissue are defined as hot nodes. The hot and positive nodes are identified as SNs by a gamma probe and fluorescence observation, respectively.

#### 5.2.2. SN Basin Dissection

SN basin dissection is performed according to the results of SN mapping. The gastric lymphatic basin is divided into five directions along the main arteries as follows: the left gastric artery area, including lymph node station Nos. 1, 3a, and 7; the right gastric artery area, including station Nos. 3b, 5, and 8a; the right gastroepiploic artery area, including station Nos. 4d and 6; the left gastroepiploic artery, including station Nos. 4sa and 4sb; and the posterior gastric artery area, including station No. 11p [72]. The harvested SNs are subjected to fluorescence imaging and a gamma probe in dissected basins on the back table and to intraoperative histological examination to diagnose lymphatic metastasis. The pathological status of the SNs and distribution of the SN basins determine the type of gastrectomy to be performed. In cases of metastasis-positive SNs, standard gastrectomy with D2 lymphadenectomy should be performed. In cases of metastasis-negative SNs, personalized optimal function-preserving gastrectomy with SN basin dissection would be possible for the individual patient as follows: local resection (LR) of the stomach, segmental gastrectomy, PPG, and PG [2,3,4]. The distribution of SN basins and the histological status of SNs would be useful in deciding the minimized extent of gastric resection and avoiding the universal application of standard gastrectomy, such as TG or DG, with D2 lymphadenectomy.

However, there are several issues with performing SNNS in clinical practice, including the learning curve, the need for radioisotope equipment and additional staff to identify sentinel nodes on the back table, and the extra load on laboratory technicians and pathologists. Therefore, currently, the implementation of SNNS would be limited to specialized facilities with well-trained surgeons.

#### 5.2.3. Local Resection of the Stomach in EGC

In cases of laparoscopic LR of the stomach, laparoscopic and endoscopic cooperative surgery (LECS) is useful [73]. However, in cases of malignancies, LECS without transluminal access seems to be important to avoid tumor seeding [74]. Because gastric cancer requires treatment using a non-exposure technique to avoid the potential risk of postoperative peritoneal dissemination, various modifications to LECSs have emerged [75,76,77]. Non-exposed endoscopic wall inversion surgery (NEWS) and CLEAN-NET (combination of laparoscopic and endoscopic approaches to neoplasia with a non-exposure technique) are examples of useful, modified LECS procedures for local full-thickness resection of the stomach without intentional perforation for EGC [75,76]. The surgical steps in NEWS are as follows: (1) mucosal and serosal markings are placed endoscopically and laparoscopically, respectively; (2) a submucosal injection is administered endoscopically, and a circumferential seromuscular incision is made laparoscopically; (3) the lesion is inverted toward the inside of the stomach by seromuscular suturing; (4) a circumferential mucosal incision is made using the technique for ESD, and (5) finally, the lesion is retrieved perorally. NEWS has the advantage of securing a surgical margin and minimizing the extent of resection because the primary lesion and mucosal markings are directly visible through the endoscope. Non-exposure simple suturing endoscopic full-thickness resection (NESS-EFTR) is also a non-exposed method of EFTR that inverts the stomach wall [78]. The difference between NESS-EFTR and NEWS is that an endoscopic mucosal incision is made to mark the dissection line followed by a simple laparoscopic seromuscular suture, which is made to invert the stomach wall and tumor. NESS-EFTR can be technically easier than NEWS. CLEAN-NET is another innovative technique for non-exposed LECS, in which the primary lesion is lifted toward the peritoneal cavity and finally resected laparoscopically, as opposed to NEWS.

### 5.3. Surgical Outcomes of SNNS for EGC

Few studies have assessed the QOL and nutritional status of patients following LR of the stomach with SNNS. According to an assessment using the PGSAS-45, patients who underwent CLEAN-NET with SNNS had better QOL than those who underwent laparoscopic distal gastrectomy (LDG) [79]. It was also reported that changes in body weight and the prognostic nutritional index were better in the LR group than in the LDG group [79]. As shown in Table 3, the change in body weight after LR ranged from 96.2% to 97.4% [69,79]. These results suggest that weight loss following LR may be less than that following other function-preserving gastrectomies such as PG (86.4–90.4%) and PPG (93.1–97.0%). However, these studies consisted of a retrospective analysis of a small sample size from a single institution, which may have resulted in bias that influenced several results.

## 6. Perspectives

Ongoing Japanese and Korean trials of SNNS are expected to verify whether limited gastrectomy with SN basin dissection achieves similar oncological outcomes and improves patients’ QOL compared to standard gastrectomy with radical lymphadenectomy for patients with EGC. In the near future, the SN concept is expected to form the basis for establishing an ideal, personalized, minimally invasive function-preserving treatment for patients with EGC that will improve the patients’ postoperative QOL without compromising long-term survival.

## 7. Conclusions

Function-preserving gastrectomy for patients with EGC appears to help maintain postoperative gastric function and improve patients’ QOL. It is also considered feasible from the perspective of oncological safety. However, most reports that support function-preserving gastrectomy are retrospective cohort studies, and the evidence is insufficient. The indications for function-preserving gastrectomy should be carefully considered to avoid the possibility of inadequate treatment.

## Figures and Tables

**Figure 1 cancers-13-06223-f001:**
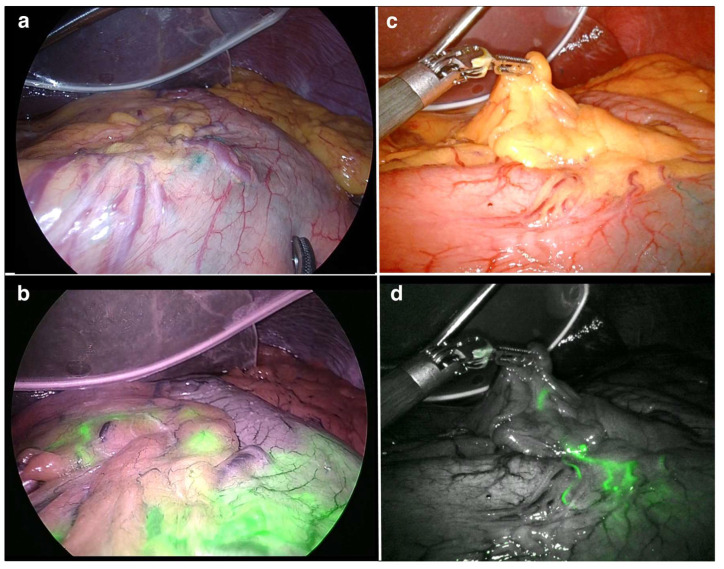
Intraoperative findings of SN mapping. (**a**) Sentinel lymphatic staining with ICG under ordinary laparoscopy; (**b**) ICG fluorescence image of the same area as in (**a**) using VISERA ELITE II system (Olympus, Japan); (**c**) Sentinel lymphatic staining with ICG under ordinary laparoscopy in another case; (**d**) ICG fluorescence image of the same area as in (**c**) using FireFly of da Vince Surgical System (Intuitive, USA). SN, sentinel node; ICG, indocyanine green.

**Table 1 cancers-13-06223-t001:** Change in body weight after proximal and total gastrectomy.

References	Year	Study Design	Institution	Sample Size (n)			Change of BW (%)		*p*-Value	WMD
				Total	PG	TG	PG	TG		
Xu [18]	2019	meta-analysis	–	816	–	–	–	–	0.000	4.33
An [35]	2008	retrospective	single	423	89	334	86.4	87.4	N.S.	
Nozaki [34]	2013	retrospective	single	151	102	49	88.0	85.0	N.S.	
Takiguchi [14]	2015	retrospective	multicenter	586	193	393	89.1	86.2	0.003	
Kosuga [30]	2015	retrospective	single	77	25	52	89.5	81.6	0.001	
Hosoda [31]	2016	retrospective	single	99	45	54	87.2	85.1	0.150	
Jung [9]	2017	retrospective	single	243	92	156	89.8	87.5	0.036	
Hayami [32]	2017	retrospective	single	90	43	47	88.0	85.0	0.003	
Nishigori [15]	2017	retrospective	single	50	16	34	89.3	83.7	0.034	
Ushimaru [17]	2018	retrospective, PSM	single	192	39	39	90.0	83.0	<0.001	
Sugiyama [33]	2018	retrospective	single	30	10	20	90.4	82.1	0.004	
Nomura [36]	2019	retrospective	single	60	30	30	89.3	84.1	0.001	

BW, body weight; WMD, weighted mean difference; PG, proximal gastrectomy; TG, total gastrectomy; N.S., no significant difference; PSM, propensity score-matched analysis.

**Table 2 cancers-13-06223-t002:** Change in body weight after pylorus-preserving and distal gastrectomy.

References	Year	Study Design	Institution	Sample Size (n)			Change of BW (%)		*p*-Value	WMD
				Total	PPG	DG	PPG	DG		
Mao [55]	2020	meta-analysis		4871	1955	2916	–	–	0.000	3.24
Tomita [59]	2003	retrospective	single	32	10	22	94.3	91.3	0.084	
Yamaguchi [60]	2004	retrospective	single	86	28	58	94.6	91.3	N.S.	
Shibata [46]	2004	prospective, RCT	multicenter	74	36	38	95.4	95.0	N.S.	
Nunobe [61]	2007	retrospective	single	397	194	203	93.9	90.2	<0.001	
Ikeguchi [62]	2010	retrospective	single	54	24	30	97.0	90.0	0.377	
Fujita [54]	2016	retrospective	multicenter	1222	313	909	93.1	92.1	0.052	
Hosoda [63]	2017	retrospective, PSM	single	99	32	32	93.1	91.8	0.450	

BW, body weight; WMD, weighted mean difference; PPG, pylorus-preserving gastrectomy; DG, distal gastrectomy; N.S., no significant difference; RCT, randomized clinical trial; PSM, propensity score-matched analysis.

**Table 3 cancers-13-06223-t003:** Change in body weight after local resection of the stomach.

References	Year	Study Design	Institution	Sample Size (n)			Change in BW (%)		*p*-Value
				Total	LR	DG	LR	DG	
Okubo [79]	2017	retrospective	single	69	25	44	97.4	93.0	0.007
Yamaguchi [69]	2019	retrospective	single	42	42		96.2		

BW, body weight; LR, local resection; DG, distal gastrectomy.

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
