# Peer review of "Function-Preserving Gastrectomy for Early Gastric Cancer"

_cancers, 2021, doi:10.3390/cancers13246223_

Round 1

Reviewer 1 Report

This review article describes the function-preserving gastrectomy for EGC very well.

#1. Most review articles describe the method of obtain data, but this article omit.

#2. Authors commented that the length of the pyloric cuff in pylorus-preserving gastrectomy (PPG) is important to prevent pyloric dysfunction. In this article, readers could misunderstand that the gastric stasis could be avoid with enough pyloric cuff only. Recent prospective study showed postoperative pyloric stenosis was significantly frequent in the PPG than DG groups (P = 0·026), (Park et al Br J Surg 2021 Sep 27;108(9):1043-1049.) Gastric stasis in patients with PPG is very important issue. Thus I recommend to describe more detailed description about gastric stasis. Whether a branch of vagus nerve is preserved or not is another cause of gastric stasis?  

#3. Authors comment NEWS and CLEAN-NET as the laparoscopic and endoscopic approach with non-exposure technique. Authors used reference the Case reports using NEWS and CLEAN-NET. Current largest trial for the laparoscopic and endoscopic approach with non-exposure technique is SENORITA3 pilot study using Non-exposure Simple Suturing Endoscopic Full-thickness Resection (NESS-EFTR) technique (Eom et al. J Gastric Cancer 2020 Sep;20(3):245-255). NESS-EFTR technique also should be commented as the laparoscopic and endoscopic approach with non-exposure technique.

Reviewer 2 Report

Thank you for giving me an opportunity to evaluate this review article entitled “Function-preserving gastrectomy for early gastric cancer”. This manuscript nicely summarized the indication, surgical strategy, and surgical outcomes of function-preserving gastrectomy such as proximal gastrectomy (PG), pylorus-preserving gastrectomy (PPG), and sentinel node navigation surgery (SNNS). This manuscript was well written, but there were some minor concerns as below.

Points

  1. PG or PPG is type of gastrectomy that necessarily reduces the extent of lymphadenectomy and gastric resection compared to standard gastrectomy. Meanwhile, SNNS is not type of gastrectomy but one of the surgical techniques that may contribute to avoiding standard gastrectomy with radical lymphadenectomy. PG or PPG is usually used as a limited surgery for early gastric cancer with some recommendation in the Japanese Gastric Cancer Treatment Guidelines whereas SNNS is not the standard of care at this time. Please clearly indicate these points.
  1. One of the interesting topics in PG is the incidence of postoperative reflux esophagitis or anastomotic stenosis in each method of reconstruction. Please show some data in the body of the manuscript or a table.
  1. Regarding the reconstruction after PG, please describe the technical tips in double tract reconstruction (DT). Are there any specific issues with DT?
  1. Although SNNS seems to be a very promising surgical concept for the treatment of early gastric cancer, please show a little more about the problems in conducting SNNS in clinical practice.
